# Retinal Nerve Fiber Layer Imaging with Two Different Spectral Domain Optical Coherence Tomographs: Normative Data for Romanian Children

**DOI:** 10.3390/diagnostics13081377

**Published:** 2023-04-10

**Authors:** Iulia-Andrada Nemeș-Drăgan, Ana-Maria Drăgan, Mădălina Claudia Hapca, Mara Oaida

**Affiliations:** 1Department of Ophthalmology, “Iuliu Hatieganu” University of Medicine and Pharmacy, 3-5 Clinicilor Str., 400006 Cluj-Napoca, Romania; 2Ophthalmology Clinic, Emergency County Hospital, 3-5 Clinicilor Str., 400006 Cluj-Napoca, Romania; 3Department of Medicine and Pharmacy, University of Oradea, 410087 Oradea, Romania; 4Doctoral School of Medicine, “Iuliu Hatieganu” University of Medicine and Pharmacy, 8, V.Babes Str., 400012 Cluj-Napoca, Romania; 5General Medicine Faculty, “Iuliu Hatieganu” University of Medicine and Pharmacy, 8, V.Babes Str., 400012 Cluj-Napoca, Romania

**Keywords:** RNFL thickness, normative data, spectral domain OCT, Romanian children

## Abstract

The purpose of this study is to analyze and compare pediatric normative data for the retinal nerve fiber layer of Romanian children using two different spectral domain optical coherence tomographs. Due to different scanning speeds and axial and transverse resolution, the results of the measurements of scans cannot be transposed. A total of 140 healthy children aged 4 to 18 were enrolled in the study. Overall, 140 eyes were scanned with a Spectralis SD-OCT (Heidelberg Technology), and the other 140 eyes were imaged with a Copernicus REVO SOCT (Optopol Technology (Zawiercie, Poland)). The mean global RNFL thickness and average RNFL thickness for the four quadrants were measured and compared. The average peripapillary RNFL thickness measured with the Spectralis was 104.03 ± 11.42 (range 81 to 126 µm), while the one measured with the Revo 80 was 127.05 ± 15.6 (range 111.43–158.28). The RNFL thickness measurements taken with the Spectralis in the superior, inferior, nasal, and temporal quadrants were 132 ±19.1, 133.5 ± 21.77, 74 ± 16.48, and 73 ± 11.95 µm, respectively, while those taken with the Revo 80 were 144.44 ± 9.25, 144.86 ±23.12, 96.49 ± 19.41, and 77 ± 11.4 µm, respectively. Multivariate analysis showed that the average RNFL thickness was not influenced by gender or eye laterality and was negatively correlated with age when we used the Spectralis device. This study provides normative data for SD-OCT peripapillary RNFL in healthy Romanian children for two different tomographs. These data help the clinician evaluate and interpret the results of optical coherence tomography for a child, taking into consideration all the technical and individual parameters.

## 1. Introduction

A quick and objective tool for diagnosis or follow-up of glaucoma in the pediatric population is of utmost importance for an ophthalmologist. Since the loss of ganglion cells axons is irreversible, prevention of this damage before it occurs should be the main concern. Perimetric and fundus changes will appear after the loss of significant axons [1]. Even if empirical formulas were used to estimate retinal ganglion cells counts from structural measurements with the aim to correlate visual field index in the assessment of neural loss from glaucoma, a linear relationship between them has not been proven [2]. Finding more objective methods to eliminate the subjectivity and improve the reproducibility of traditional clinical diagnostic parameters is always a large concern [3].

Optical coherence tomography (OCT) imaging provides in vivo histology of the anterior visual pathway and, respectively, allows reproducible quantification of retinal nerve fiber layers by segmentation [4,5,6]. It is a non-invasive test for both macula and nerve diseases that uses low-coherence light reflected by the ocular tissues to obtain high-resolution images [7].

Pediatric OCT imaging is a very intensely studied topic that is revolutionizing the diagnosis and monitoring of the pediatric optic nerve or retinal disease. The analysis of a pediatric examination is a complex and challenging task. The lack of normative data makes the diagnosis and the therapeutic decision more difficult [8].

The new-generation, high-speed imaging technology spectral-domain OCT (SD-OCT) uses a Fourier-domain interferometric method to provide up to a 5 µm resolution [9]. Several manufacturers have proposed different SD-OCTs that provide better quality and shorter image acquisition with fewer artifacts, which are characteristics that recommend this investigation for the pediatric population. The RNFL measurements taken with an earlier OCT model, TD-OCT, proved a good diagnostic ability and accuracy in glaucoma or other optic nerve pathologies for both adults and children [10,11,12,13,14,15]. Although the results of these early and late technologies are comparable, due to different scanning speed and axial and transverse resolution, the results of these measurements cannot be transposed because scanning protocols are different even among SD-OCT instruments [15,16].

The current technology used for young patients means that sometimes, we need to sacrifice better resolution for higher speed, especially in non-compliant children. Therefore, when possible, machine parameters should be adjusted for shorter scan lengths, fewer repetitions, and lower line scan density to be sure we obtain some accurate images.

Data regarding pediatric normative values for the Spectralis SD-OCT (Heidelberg Technology) are scarce, but there is also paucity of literature regarding the Copernicus REVO (Optopol Technology).

The most recent SD-OCTs expand their clinical application, but there is still a need for normative data regarding children.

The purpose of this study was to identify normative values for SD-OCT measurements of RNFL thickness of healthy eyes in normal children using two of the most used technologies in Romania and to determine the difference between RNFL parameters measured by different types of OCT.

## 2. Materials and Methods

### 2.1. Study Design and Subjects

This was an analytical prospective study carried out in two clinics: the Department of Ophthalmology, “Iuliu Hatieganu”, University of Medicine and Pharmacy, Emergency County Hospital in Cluj-Napoca, Romania, and Doctorlens Eye Clinic in Cluj-Napoca between 1 January 2021 and 31 December 2022. All procedures in the study were carried out in accordance with the Declaration of Helsinki. The legally responsible party received detailed information about the nature of the investigation and provided written informed consent before study enrollment.

The inclusion criteria included Caucasian children aged 4 to 18 with best-corrected visual acuity (BCVA) of 1 (on the Snellen visual acuity scale), a maximal difference of one line of vision between the visual acuity (VA) of both eyes, and no systemic or ocular problems other than low or medium refractive errors.

The exclusion criteria were the following: prematurity, high refractive error, defined as SE exceeding ±6 diopters (D), history of manifest strabismus or amblyopia, family history of glaucoma, any retinal or optic disc anomaly, previous ocular trauma, or delayed psychomotor development.

All children underwent complete ophthalmological examinations. The best-corrected distance VA of each eye was measured using the Snellen chart with and without correction. Refractive errors were assessed under cycloplegia using Tropicamide 1% eyedrops instilled three times, 10 min apart, and measured with an auto-refractometer (Topcon KR-8900). Intraocular pressure assessment, motility examination, stereoacuity testing (Lang test), slit lamp exam, and dilated fundoscopy were performed.

### 2.2. Spectral-Domain OCT Measurements

OCT imaging was performed with a Spectralis SD-OCT (Heidelberg Engineering (Port Melbourne, Australia), Heidelberg Germany software) and with SD-OCT Copernicus REVO (Optopol Technology, Software Version 9.7) by the same well-trained technician. All Spectralis measurements were made with preselected small eye length. Regarding the quality of the examinations, we took into consideration only well-focused images with signal strength better than 20 (40 maximum) for Spectralis and better than 10 (30 maximum) for Copernicus, as suggested by the manufacturer, with no overt misalignment or decentration of the measurement circle location around the optic disc.

All examinations were performed after pupillary dilatation with Spectralis OCT (version 7.0), which has an acquisition rate of 40,000 A lines per second, an axial resolution of 7 µm in tissue, with an 870 nm superluminescent diode source. The circular scan mode was used for peripapillary RNFL thickness measurements. For the Spectralis, a scan circle with a diameter of 3.45 mm was manually positioned to locate the optic disc at the center of the circle. For all patients, the tracking eye system and internal target was activated. Spectralis technology provides an automatic real-time function with an eye-tracking system with the purpose of increasing the image quality. With this function activated, multiple frames of the same scanning location are obtained. The data obtained are then averaged for noise reduction, and eye motion artefacts are reduced. The software calculates the average RNFL thickness overall, globally (G, 360 degrees), for the four quadrants (superior (S), inferior (I), nasal (N), and temporal (T), each 90 degrees), and then four additional sectors (superior-temporal, superior-nasal, inferior-nasal, and inferior-temporal).

The SOCT Copernicus REVO (Optopol technology) uses a very child-friendly technology that includes a real-time hardware eye-tracking function that compensates for blinking, loss of fixation, and involuntary eye moving during the OCT scanning. For compliant children, automatic scanning of the optic nerve was used. There were just a few patients that needed the manual scan after several trials with the automatic mode. The scan measurement of the optic disc and TSINT region was analyzed based on the captured OCT image of the optic disc in the [Disc 3D] mode. The RNFL profile indicates the thickness of locations through which a measurement circle (diameter 3.45 mm centering on the optic disc) passes, and the RNFL grid indicates the thickness of the region inside the measurement circle. The values of RNFL analysis are shown in a map (Figure 1b). The NFL thickness map shows thickness of the NFL layer on scanned area. The map has rings around the disc, where the RNFL thickness data are used for TSNIT analysis.

### 2.3. Statistical Analysis

Descriptive statistics were expressed using both counts and percentages for the categorical variables while using means, medians, and standard deviations for the numerical variables. Analysis of distribution was performed using Kolmogorov–Smirnov and Shapiro–Wilk tests for normality in conjunction with the derived Q-Q plots and histograms. Means (or medians) comparison was performed by using either independent samples *t*-test, Mann–Whitney U, Kruskal–Wallis H (with Dunn-Bonferroni test in post hoc), or Wilcoxon signed-rank test, each by case. Furthermore, linear relationships between scalar variables were tested using either Pearson’s or Spearman’s correlations depending on the observed distribution. With regard to categorical variables, crosstabs were created, with subsequent testing for independence using chi-square/Fisher’s exact tests depending on the counts.

Microsoft Office Excel (Office 365 Suite) was used to create the databases. All statistical analysis was performed using IBM SPSS Statistics v.26, and 95% confidence intervals are reported as indicated. The threshold of statistical significance was set at a Pearson’s *p* = 0.05.

## 3. Results

### 3.1. Study Lot

A total of 75 children were analyzed using the Spectralis SD-OCT. Due to suboptimal imaging quality, five subjects were withdrawn. A total of 140 eyes from 70 subjects met the inclusion criteria, summing 36 (51.40%) females and 34 (48.60%) males. The mean subject age (±SD) was 10.21 ± 8.32 years (range 4 to 18 years). Of the additional 75 children examined using Revo80 SD-OCT, 2 were uncooperative, and 3 were excluded due to poor image quality. Among these 140 eyes from 70 subjects admitted in the study, 43 (61.40%) were female, and 27 (38.60%) were male. The mean age (±SD) was 9.11 ± 7.20 years (range 4 to 18 years).

### 3.2. Statistical Power of the Study

We determined the statistical power of this study using between-subjects effects testing, taking into account every eye as separated from both groups. The *p*-value revealed high statistical significance for all parameters except temporal RNFL mean values. The observed power showed high values, with temporal RNFL making the exception. All values are displayed in Table 1.

### 3.3. Global and Sectoral RNFL Thickness Values

Regarding the Spectralis measurements, the mean ± SD for global RNFL thickness was 104.03 ± 22.84 μm (with a median of 103.50 μm). The inferior sector measured the highest mean, i.e., 134.64 ± 43.54 μm, and the thinnest mean was found in the nasal sector, i.e., 74.20 ± 32.96 μm. The values measured by Revo 80 were higher (*p* < 0.001) except for the temporal sector (*p* = 0.071). The mean for global RNFL thickness was 127.05 ± 31.22 μm, and the median was 127.00 μm. Concerning sectorial values, the highest mean was linked to the inferior sector, i.e., 144.86 ± 38.50 μm, although the thinnest mean RNFL appeared in the temporal sector, i.e., 77.00 ± 22.80 μm as was presented in Table 2. The mean values for both Spectralis and Revo 80 measurements are shown in Figure 2 and Figure 3.

### 3.4. Side-To-Side Differences

Going further, we investigated the influence of eye laterality on our parameters. Global RNFL thickness did not register significant differences (*p* = 0.306), with 103.31 ± 18.56 μm, 95% CI (101.10–105.53) mean value in the right eyes and 104.74 ± 26.50 μm, 95% CI (101.58–107.90) in the left eyes on the Spectralis measurements, respectively. In regard to sectorial variability, superior and inferior RNFL means revealed higher values in the left eye (Table 3).

Likewise, the Revo80 recorded no significant differences between the left and the right eyes in global RNFL values (*p* = 1.000). The means were 126.63 ± 30.38 μm, 95% CI (123.01–130.25) for right eyes and 127.47 ± 32.24 μm, 95% CI (123.63–131.32) for left eyes. Association with laterality was only relevant for nasal RNFL sector (*p* = 0.023), which showed differences between the eyes, with higher values in the right; the means were 98.43 ± 40.90 μm, 95% CI (93.55–103.31) and 94.56 ± 36.52 μm, 95% CI (90.20–98.91), respectively. (Table 4)

### 3.5. Gender Influence on Global and Sectorial RNFL Thickness

The mean value for global RNFL was 103.69 ± 15.54 μm, 95% CI (101.81–105.57) in males and 104.35 ± 28.16 μm, 95% CI = (101.04, 107.66) in females for the Spectralis, respectively, and 126.15 ± 28.32 μm, 95% CI = (122.28, 130.01) in males and 127.62 ± 33.02 μm, 95%CI = (124.07, 131.16) in females for the Revo80 (Table 5). There were no differences found between genders regarding global (*p* = 0.562 in Spectralis; *p* = 0.772 in Revo80) or sectorial RNFL thickness values. Higher values of RNFL readings when measured with the Revo80 remained consistent when filtering by gender except for the temporal sector RNFL values (*p* = 0.125 for males and *p* = 0.196 for females).

### 3.6. Age and Age Groups Variability

Age was negatively correlated with global (*p* = 0.039, Spearman’s rho = −0.175) and superior sector RNFL values (*p* = 0.042, Spearman’s rho = −0.172) on the Spectralis measurements (Figure 4). The Revo80 revealed negative correlation between age and temporal sector RNFL values (*p* = 0.024, Spearman’s rho = −0.191).

We defined three age groups: group 1 includes children aged 4 to 9, group 2 includes children from 9 to 14 years, and group 3 includes children from 14 to 18 years (Table 6). The Kruskal–Wallis H test was used to compare RNFL values between those three age groups. In the case of Spectralis measurements, differences between the values of global RNFL thickness (*p* < 0.001) and superior (*p* = 0.001), inferior (*p* = 0.017), and temporal (*p* = 0.045) sectors were found. Post hoc, we used the Dunn–Bonferroni test to further compare RNFL values between the age groups taken two by two. For global RNFL thickness, we found differences between groups 1 and 3 (*p* = 0.013) and groups 2 and 3 (*p* < 0.001). Between groups 1 and 3 (*p* = 0.006) and groups 2 and 3 (*p* = 0.001), the superior sector RNFL thickness registered statistically significant differences. Differences were found only between groups 2 and 3 when we analyzed inferior (*p* = 0.014) and temporal (*p* = 0.040) sectors. Regarding Revo80 measurements, differences were only found in the temporal sector (Kruskal–Wallis *p* = 0.048), with group 1 registering statistically significantly higher values than group 3 (Dunn–Bonferroni *p* = 0.041).

### 3.7. ISNT Rule

The ISNT rule is defined by descending values for pRNFL thickness sectors in the following order: inferior > superior > nasal > temporal. We investigated the presence and prevalence of ISNT rule in our groups, taking into consideration the eye laterality with different groups for right and left eyes. Further, we evaluated the rule prevalence without regard for laterality in all 140 eyes for each OCT model. In the Spectralis group, 47.1%, and 34.3%, respectively, respected the rule when evaluating the right eye and left eye separately. The global prevalence in all 140 eyes was 40.7%. Regarding the Revo80 measurements, prevalence of ISNT rule was 27.1% in the right eye and 24.3% in the left eye. When both eyes were evaluated as a group, 25.7% followed the rule criteria.

We further observed the correlation between the ISNT rule and global RNFL thickness. In the Spectralis measurements, the global RNFL values were higher (*p* = 0.007) in the eyes following the rule (mean = 108.56 ± 26.37 μm) as opposed to the ones that did not (mean = 102.46 ± 20.73 μm). The Revo80 measurements were higher in eyes following the rule as well (*p* = 0.023). The mean was 131.21 ± 35.41 μm in the eyes that followed the rule in contrast to the eyes that did not respect the criteria, where the mean was 124.19 ± 26.75 μm.

## 4. Discussion

Nowadays, the OCT is an indispensable tool for diagnosis and follow-up of optical neuropathies and glaucoma in adults. Many studies demonstrate its feasibility and reproducibility for pediatric age and an increase in diagnostic accuracy if the examiner had normative data for the device that is used [17,18,19,20,21]. SD-OCT is a fast, easy, and noninvasive examination that allows the clinician to quantitatively measure the thickness of the retinal nerve fiber layer.

None of the available technologies and software have incorporated a normative data base for pediatric population; thus, these machines use adult-normative data to compare the measurement ranges and parameters.

The current study reported normative values of RNFL thickness using two different technologies in 140 healthy Caucasian children aged 4–18 years. The RNFL measurements obtained may vary according to the SD-OCT used and are not transposable.

### 4.1. Comparison of Global and Sectoral RNFL Thickness Values

The mean global RNFL thickness after Spectralis measurements was 104.03 ± 22.84 μm in our group study. This value was similar to other studies performed with SD-OCT on Caucasians [22,23] or in other races aged 4–18 years [24,25,26]. The “double-hump” pattern was conformed, with the RNFL’s highest thickness in the inferior quadrant, followed by the superior quadrant and with very similar values for the nasal and temporal quadrants [24,25,26,27,28,29].

In comparison with Spectralis, the Copernicus technology, REVO80, measured significantly higher mean global RNFL values, in accordance with the single literature result performed on the same age group and same conditions [30].

Silverman et al. concluded in a study on an adult population that two different reference databases analyzed from two different technologies showed moderate to substantial agreement, mentioning that both had a high specificity for identifying healthy eyes [31].

### 4.2. Comparison of Eye Laterality in Other Studies

The global RNFL thickness presented no significant differences between the right and left eye when we measured with either Spectralis or Copernicus, which are results in accordance with other studied measurements with the same technology [21,32,33,34]. Sectorial measurements show no significant difference between the eyes when performed with Spectralis, whereas Copernicus scans show slightly different thicknesses in the right eye in the nasal quadrant. There are few studies that report higher values of RNFL parameters, but these occur in the left eye [13,18,35]. The interocular differences were also studied by Altemir et al. [36] and Tan et al. [37], who recommended that RNFL thickness should not be more than 13 μm.

### 4.3. Comparison of Gender Influence on Global and Sectorial RNFL Thickness with Other Studies

Our study did not find statistically significant differences between male and female RNFL thickness, in accordance with an important number of studies regardless of the technology that was used [17,18,21,24,25,30,33,34,38,39,40,41,42,43]. Turk et al. found a significant difference between female and male children in terms of RNFL only in the temporal inferior segment, without any significant difference in the other RNFL parameters [22]. In a Zhu et al. study, girls had higher thickness in the inferior and temporal RNFL sectors [44]. Huynh et al. and Chen et al. reported greater values in global RNFL, inferior sector, and temporal sectors in boys [45,46].

### 4.4. Comparison of Age Group Variability with Other Studies

The influence of age on RNFL thickness and OCT parameters was investigated in several studies. Our study reports a negative correlation between age and RNFL thickness with both scanning technologies that we used, in accordance with Yanni et al. [25] and Salchow et al. [21] in a similar age group. A recent review also concluded that there are age-related variations [7]. Goh et al. reported a negative correlation between superior sector RNFL thickness and age [47]. Lee et al. [27] and Ali et al. demonstrated a statistically significant effect of age on RNFL thickness [48]. In a recent study, Xiu-Juan Zhang et al. noted that p-RNFL increases significantly with age among children 6 to 8 years old, similar to the results that we obtained using Copernicus technology among the same age group.

However, most studies reported no significant association between age and RNFL thickness in the pediatric population [17,22,23,24,25,26,28,30,32,33,34,35,38,39,40,41,44,49,50,51,52,53].

The impact of age on RNFL thickness in adults was explained in a recent meta-analysis as an age-related loss of several parameters [54], whereas in the pediatric population, the authors did not find a correlation between age and RNFL thickness. Hong et al. reported the risk of overestimating the RNFL thickness in children < 15 years if the ocular magnification effects are not taken into consideration [55].

### 4.5. Comparison of ISNT Rule on OCT Parameters with Other Studies

The neuroretinal rim (NRR) is an extension of the intrapapillary part of the optic nerve fibers. NRR thinning is a sign of glaucoma; therefore, evaluation of its configuration is of utmost importance [56]. It is known that glaucoma patients do not respect the ISNT rule. Studies performed on pediatric populations demonstrate high variability and discordance, so the disobedience of the ISNT rule cannot be interpreted as abnormal [54]. Pogrebniak demonstrated that disobedience of ISNT rules occurs more often in pediatric populations with large optic disc cups of nonglaucomatose origin [56]. Prematurity may also have an influence on disobeying the rule criteria based on its impact on disc morphology, as it was observed in children born prematurely (<32 gestational weeks) [56].

The majority of the pediatric cohorts respect the ISNT rule [17,18,21,33,34,35,40]. However, the violation of the ISNT can be observed in healthy eyes, and upon conducting further investigations, it should not be classified as pathological [57].

### 4.6. Strengths and Limitations

There is a scarcity of literature on the Copernicus technology and pediatric populations. The strength of this study is that it provides normative data for two different devices for children. The comparison that was made emphasizes once again the need to use the same device for the follow-up of a pediatric patient.

The normative data presented in this study can help clinicians manage optic nerve pathologies, glaucoma, or risk factors in children. By comparing individual OCT scanning technology results with normative data, technical and individual parameters can be interpreted better and be of clinical use. Pediatric-normative parameters are usually not provided by most companies, making routine pediatric ophthalmic practice deficient in this regard. The article presents clinically important comparisons of different parameters based on instrument, eye laterality, gender, age group, and ISNT rule.

There are also some limitations. The first one regards our sample size, which can be classified as limited. Second, axial length was not taken into account when we used the Copernicus technology, whereas the Spectralis examinations were all made with preselected small axial length. However, there is a great variability of results reported on the importance of this factor [21,42,58,59].

Another limit could be that we analyzed only European Caucasian children. There are studies that suggest differences between races. Caucasian, Black (Afro Americans), and Asian populations have been mostly studied until now [48,52,60,61].

## 5. Conclusions

This study provides useful normative data for two of the most used technologies in Romania. Our results will help clinicians interpret and follow-up the pediatric pathology that affects or could affect the optic nerve. The comparison emphasizes again that the results obtained with different technologies are not interchangeable.

## Figures and Tables

**Figure 1 diagnostics-13-01377-f001:**
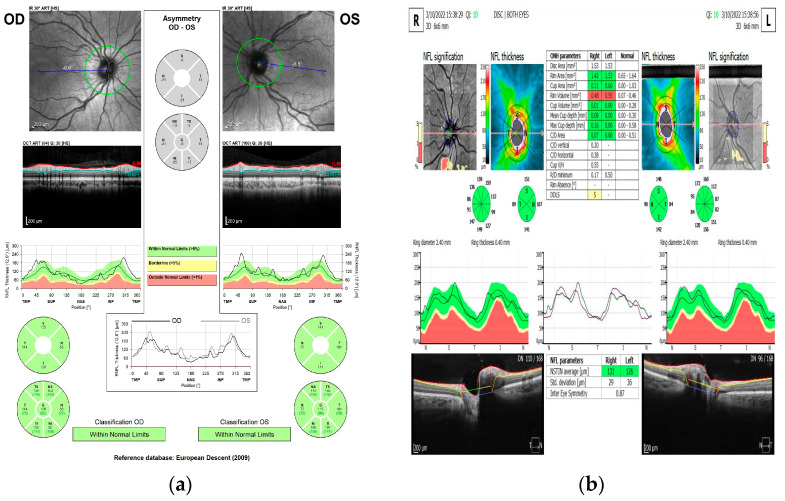
(**a**) Spectralis SD-OCT scans of retinal nerve fiber layer thickness map. (**b**) SOCT Copernicus REVO NFL thickness profile. [Personal database].

**Figure 2 diagnostics-13-01377-f002:**
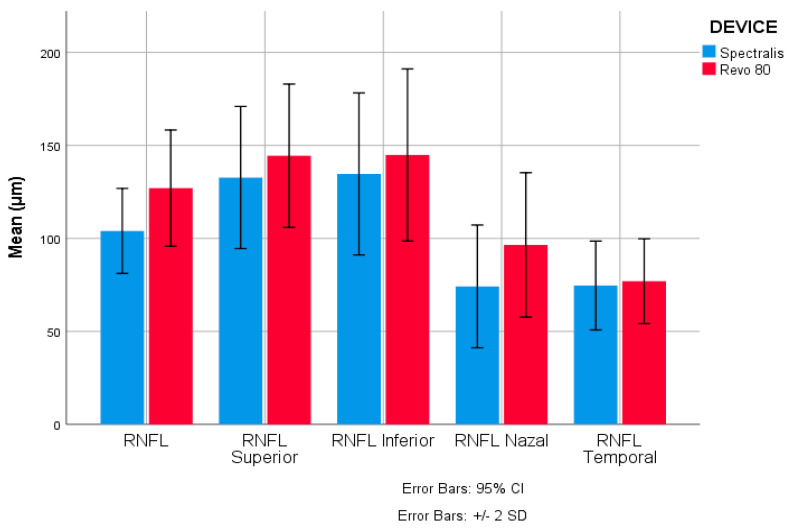
Clustered bar chart displaying the differences between mean values of global and sectorial RNFL in Spectralis and Revo80.

**Figure 3 diagnostics-13-01377-f003:**
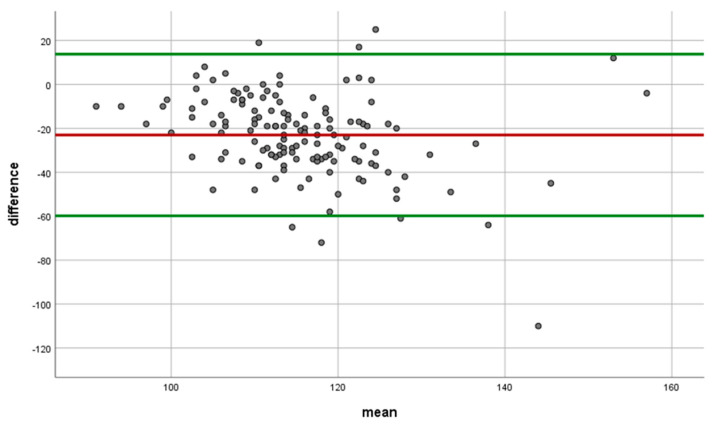
Bland–Altman chart of Spectralis and Revo80 measurements for Global RNFL, where ◦ is difference, **--** mean, and **--** lower and upper 95% limit values.

**Figure 4 diagnostics-13-01377-f004:**
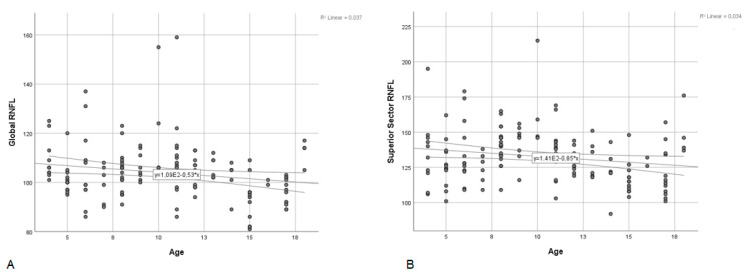
Scatterplot of global RNFL (**A**) and superior sector RNFL (**B**) thickness as a function of age of the subjects on Spectralis measurements.

**Table 1 diagnostics-13-01377-t001:** *p*-Values and observed power for global and sectorial RNFL thickness values.

RNFL	*p*-Value	Observed Power
Global	<0.001	>0.999
Superior	<0.001	0.999
Inferior	<0.001	0.967
Nasal	<0.001	>0.999
Temporal	0.096	0.383

**Table 2 diagnostics-13-01377-t002:** Descriptive values of global and sectorial RNFL thickness in the Spectralis and Revo 80.

RNFL, μm	Spectralis					Revo 80					*p*-Value
	Mean (SD)	Median	Range	95% CI		Mean (SD)	Median	Range	95% CI		
Global	104.03 (11.42)	103.50	81–159	102.12	105.94	127.05 (15.61)	127.00	96–199	124.44	129.66	<0.001
Superior	132.73 (19.10)	132.00	92–215	129.54	135.92	144.44 (19.25)	145.00	113–239	141.23	147.66	<0.001
Inferior	134.64 (21.77)	133.50	92–226	131.00	138.28	144.86 (23.12)	145.00	66–255	140.99	148.72	<0.001
Nasal	74.20 (16.48)	74.00	42–168	71.44	76.96	96.49 (19.41)	95.00	59–158	93.25	99.74	<0.001
Temporal	74.67 (11.95)	73.00	52–120	72.67	76.67	77.00 (11.40)	76.00	54–108	75.09	78.91	0.071

**Table 3 diagnostics-13-01377-t003:** Global and sectorial RNFL thickness means filtered by eye laterality on the Spectralis measurements.

	Right Eye		Left Eye		
RNFL, μm	Mean (SD)	Range	Mean (SD)	Range	*p*-Value
Global	103.31 (9.28)	81–137	104.74 (13.25)	82–159	0.306
Superior	130.67 (17.03)	101–179	134.79 (20.88)	92–215	0.011
Inferior	132.43 (18.25)	92–196	136.86 (24.73)	96–226	0.044
Nasal	74.39 (13.86)	42–107	74.01 (18.85)	47–168	0.474
Temporal	75.50 (12.98)	52–120	73.84 (10.85)	54–107	0.306

**Table 4 diagnostics-13-01377-t004:** Global and sectorial RNFL thickness means filtered by eye laterality on Revo80 measurements.

	Right Eye		Left Eye		
RNFL, μm	Mean (SD)	Range	Mean (SD)	Range	*p*-Value
Global	126.63 (15.19)	96–170	127.47 (16.12)	99–199	1.000
Superior	143.33 (17.18)	114–196	145.56 (21.19)	113–239	0.051
Inferior	146.20 (19.97)	103–210	143.51 (25.97)	66–255	0.609
Nasal	98.43 (20.45)	59–151	94.56 (18.26)	64–158	0.023
Temporal	75.97 (10.47)	54–108	78.03 (12.24)	56–105	0.313

**Table 5 diagnostics-13-01377-t005:** Means (filtered by gender) for global and sectorial RNFL thickness.

	Spectralis			Revo80			*p*-Value	
RNFL, μm Mean (SD)	Male	Female	*p*-Value	Male	Female	*p*-Value	Male	Female
N	68	72		54	86			
Global	103.69 (7.77)	104.35 (14.08)	0.562	126.15 (14.16)	127.62 (16.51)	0.772	<0.001	<0.001
Superior	133.32 (15.93)	132.17 (21.76)	0.427	142.87 (17.47)	145.43 (20.33)	0.489	0.002	<0.001
Inferior	133.28 (15.96)	135.93 (26.15)	0.813	142.67 (23.97)	146.23 (22.60)	0.998	0.015	<0.001
Nasal	73.57 (13.35)	74.79 (19.06)	0.921	94.31 (20.49)	97.86 (18.70)	0.250	<0.001	<0.001
Temporal	74.79 (9.54)	74.56 (13.91)	0.460	77.67 (10.96)	76.58 (11.71)	0.254	0.125	0.196

**Table 6 diagnostics-13-01377-t006:** Age group means for Global and Sectorial RNFL thickness.

	*n*		Global RNFL Mean (SD), μm		Superior RNFL Mean (SD), μm		Inferior RNFL Mean (SD), μm		Nasal RNFL Mean (SD), μm		Temporal RNFL Mean (SD), μm	
Age Groups	Spectralis	Revo80	Spectralis	Revo80	Spectralis	Revo80	Spectralis	Revo80	Spectralis	Revo80	Spectralis	Revo80
4–9	60	72	104.53 (9.98)	128.42 (16.77)	134.83 (18.78)	145.94 (21.06)	134.90 (22.14)	143.90 (27.34)	74.03 (13.08)	96.99 (17.86)	74.35 (10.26)	78.53 (11.66)
9–14	44	48	108.23 (13.09)	126.81 (12.65)	137.50 (18.47)	143.96 (16.03)	140.70 (23.83)	146.92 (15.49)	77.48 (21.80)	97.04 (20.19)	77.50 (12.65)	77.00 (10.80)
14–18	36	20	98.06 (8.96)	122.70 (17.57)	123.39 (17.56)	140.20 (19.74)	126.81 (15.77)	143.35 (22.50)	70.47 (13.39)	93.40 (23.35)	71.75 (13.18)	71.50 (10.64)

## Data Availability

The data presented in this study are available on request to the corresponding author.

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
