# Peer review of "Retinal Nerve Fiber Layer Imaging with Two Different Spectral Domain Optical Coherence Tomographs: Normative Data for Romanian Children"

_diagnostics, 2023, doi:10.3390/diagnostics13081377_

Round 1

Reviewer 1 Report

Thank you for the opportunity to review.

It's an interesting study, but I think the discussion should include what kind of clinical implications this result brings.

minor comment

Is L107 SOOCT a mistake?

Author Response

  1. It's an interesting study, but I think the discussion should include what kind of clinical implications this result brings.

Thank you for your review. I added in discussion an paragraph about the clinical implications of this study (line 365) : The normative data presented in this study can help clinicians manage optic nerve pathologies, glaucoma, or risk factors in children. By comparing individual OCT scanning technology results with normative data, technical and individual parameters can be interpreted better and be of clinical use. Pediatric normative parameters are usually not provided by most companies, making routine pediatric ophthalmic practice deficient in this regard. The article presents clinically important comparisons of different parameters based on instrument, eye laterality, gender, age group, and ISNT rule.

  1. Is L107 SOOCT a mistake?

Yes, I changed in SD-OCT.

Reviewer 2 Report

This manuscript tried to analyze and compare pediatric normative data for the retinal nerve fiber layer of Romanian children using two different spectral domain optical coherence tomographs. Overall, the conception of the study design is well documented. In fact, I don't have much to add to the study design or the construction of the paper. But there are still some minor grammar error needed to edit. Please have some native English speaker revise it to make it easier to read.

Specifically, the authors should fix the following problems which will help the work achieve the published criteria of this journal.

1.     Please describe the statistical power of your research.

2.     Please mention that how to ensure high quality images from the child in group 1. By the way, please explain why do not use the same study subjects between these two devices. As we can see in Table 5, the numbers between groups are different.

3.     It is much better to use the Bland-Altman plot to determine the 95% limit of agreement between the devices.

4.     For Figure 1, it is much better to yield the high-resolution images from the device other than taking a picture from paper-copy.

5.     In line 223, Figure 1 should be changed as Figure 2. In line 227, Figure 2 should be changed as Figure 3. By the way, the pattern of Figure 3 is not very well. Please improve it.

Author Response

 Thank you very much for your comments.

  1. Please describe the statistical power of your research:

I added the 3.2 subchapter : 3.2. Statistical power of the study

We determined the statistical power of this study using Between-Subjects Effects testing taking into account every eye as separated from both groups. The p value revealed high statistical significance for all parameters except temporal RNFL mean values. The observed power showed high values, temporal RNFL making the exception. All values are displayed in Table 1.

Table 1. p values and Observed power for Global and Sectorial RNFL thickness value  s

RNFL

P value

0bserved power

Global

<0.001

>0.999

Superior

<0.001

0.999

Inferior

<0.001

0.967

Nasal

<0.001

>0.999

Temporal

0.096

0.383

  1. Please mention that how to ensure high quality images from the child in group 1. By the way, please explain why do not use the same study subjects between these two devices. As we can see in Table 5, the numbers between groups are different.

The number between groups is the 140 eyes measured with Spectralis and 140 eyes measured with Copernicus. We examined the subjetcs in two different centers from where we recruited the patients, unfortunately the subjectes were different but all of them were eligible.

  1. It is much better to use the Bland-Altman plot to determine the 95% limit of agreement between the devices.

Bland-Altman chart of Spectralis and Revo80 measurements for Global RNFL

  1. For Figure 1, it is much better to yield the high-resolution images from the device other than taking a picture from paper-copy.

5.In line 223, Figure 1 should be changed as Figure 2. In line 227, Figure 2 should be changed as Figure 3. By the way, the pattern of Figure 3 is not very well. Please improve it.

I removed Figure 3.

  1. Please describe the statistical power of your research:

 Thank you very much for your comments.

I added the 3.2 subchapter : 3.2. Statistical power of the study

We determined the statistical power of this study using Between-Subjects Effects testing taking into account every eye as separated from both groups. The p value revealed high statistical significance for all parameters except temporal RNFL mean values. The observed power showed high values, temporal RNFL making the exception. All values are displayed in Table 1.

Table 1. p values and Observed power for Global and Sectorial RNFL thickness value  s

RNFL

P value

0bserved power

Global

<0.001

>0.999

Superior

<0.001

0.999

Inferior

<0.001

0.967

Nasal

<0.001

>0.999

Temporal

0.096

0.383

  1. Please mention that how to ensure high quality images from the child in group 1. By the way, please explain why do not use the same study subjects between these two devices. As we can see in Table 5, the numbers between groups are different.

The number between groups is the 140 eyes measured with Spectralis and 140 eyes measured with Copernicus. We examined the subjetcs in two different centers from where we recruited the patients, unfortunately the subjectes were different but all of them were eligible.

  1. It is much better to use the Bland-Altman plot to determine the 95% limit of agreement between the devices.

Bland-Altman chart of Spectralis and Revo80 measurements for Global RNFL

  1. For Figure 1, it is much better to yield the high-resolution images from the device other than taking a picture from paper-copy.

5.In line 223, Figure 1 should be changed as Figure 2. In line 227, Figure 2 should be changed as Figure 3. By the way, the pattern of Figure 3 is not very well. Please improve it.

I removed Figure 3.

Reviewer 3 Report

This manuscript is an excellent description of measurements of the peripapillary thickness of the retinal nerve fiber layer (pRNFL) done by two different optical coherent tomographs (OCT) [Spectralis and Copernicus] in 140 healthy Caucasian children aged 4 to 18 years. The values of the average thickness and of the thickness of 4 quadrant were different between these 2 devices – and not influenced by age or eye laterality. Therefore, the results are not interchangeable. This data helps the clinician evaluate and interpret the results of optical coherence tomography for children of this age group. A perfect new data base!

Some remarks:

Line 161: please change patients to subjects

Line 258: “The ISNT rule is defined as descending values for RNFL thickness sectors….” Per definition it is not the RNFL thickness per se, it is the width of the rim shape (widest inferiorly, etc.). This corresponds with the thickness of the pRNFL. [RNFL thickness is a vertical parameter, ISNT rule depends on a horizontal parameter]. Please adapt.

Author Response

Thank very much for your remarks:

  1. Line 161: please change patients to subjects

Due to suboptimal imaging quality, 5 subjects were withdrawn. 140 eyes from 70 subjects have met the…

  1. Line 258: “The ISNT rule is defined as descending values for RNFL thickness sectors….” Per definition it is not the RNFL thickness per se, it is the width of the rim shape(widest inferiorly, etc.). This corresponds with the thickness of the pRNFL. [RNFL thickness is a vertical parameter, ISNT rule depends on a horizontal parameter]. Please adapt.

The ISNT rule is defined as descending values for pRNFL thickness sectors in the following order: inferior>superior>nasal>temporal.

Reviewer 4 Report

The aim of the study titled " Retinal nerve fiber layer imaging with two different spectral domain optical coherence tomographs: normative data for Romanian children" is to establish normative data for SD-OCT peripapillary RNFL in healthy Romanian children using two different tomographs.

Although OCTs are commonly used for managing retinopathies and glaucoma in adults, their use in pediatrics is limited. The normative data presented in this study can help clinicians manage optic nerve pathologies, glaucoma, or risk factors in children. By comparing individual OCT scanning technology results with normative data, technical and individual parameters can be interpreted better and be of clinical use. Pediatric normative parameters are usually not provided by most companies, making routine pediatric ophthalmic practice deficient in this regard. The article presents clinically important comparisons of different parameters based on instrument, eye laterality, gender, age group, and ISNT rule.

The study is well-organized with clear headings and subheadings that aid in structuring the content and linking specific topics. Each parameter is highlighted, emphasizing its importance. To aid pediatric ophthalmologists, the authors could provide a flow chart on the clinical management of children based on OCT results.

Overall, the study is well-written and clinically relevant. The figures and tables are relevant, and descriptive, and help convey the results. The references are suitable, and the topic is intriguing. However, a native English doctor's editing can improve the text's English and flow.

Author Response

Thank you very much for your suggestions.

I used your suggestion to improve our disscutions: The normative data presented in this study can help clinicians manage optic nerve pathologies, glaucoma, or risk factors in children. By comparing individual OCT scanning technology results with normative data, technical and individual parameters can be interpreted better and be of clinical use. Pediatric normative parameters are usually not provided by most companies, making routine pediatric ophthalmic practice deficient in this regard. The article presents clinically important comparisons of different parameters based on instrument, eye laterality, gender, age group, and ISNT rule.

To aid pediatric ophthalmologists, the authors could provide a flow chart on the clinical management of children based on OCT results.

Our study aims to provide pediatric population data for the purpose of guidance for creating future normative data base and clinical protocols based on OCT results. We are not in the position to advise any clinical management for now.
